# Genome mining for macrolactam-encoding gene clusters allowed for the network-guided isolation of β-amino acid-containing cyclic derivatives and heterologous production of ciromicin A

Elena Seibel[1,2], Soohyun Um[1,3], Marie Dayras [2], Kasun H. Bodawatta[4,5], Martinus de Kruijff [2], Knud A. Jønsson [5,6], Michael Poulsen [7], Ki Hyun Kim [8✉] & Christine Beemelmanns [1,2,9✉]

β-Amino acid-containing macrolactams represent a structurally diverse group of bioactive natural products derived from polyketides; however we are currently lacking a comprehensive overview about their abundance across bacterial families and the underlying biosynthetic diversity. In this study, we employed a targeted β-amino acid-specific homology-based multi-query search to identify potential bacterial macrolactam producers. Here we demonstrate that approximately 10% of each of the identified actinobacterial genera harbor a biosynthetic gene cluster (BGC) encoding macrolactam production. Based on our comparative study, we propose that mutations occurring in specific regions of polyketide synthases (PKS) are the primary drivers behind the variation in macrolactam ring sizes. We successfully validated two producers of ciromicin A from the genus *Amycolatopsis*, revised the composition of the biosynthetic gene cluster region *mte* of macrotermycins, and confirmed the ciromicin biosynthetic pathway through heterologous expression. Additionally, network-based metabolomic analysis uncovered three previously unreported macrotermycin congeners from *Amycolatopsis* sp. M39. The combination of targeted mining and network-based analysis serves as a powerful tool for identifying macrolactam producers and our studies will catalyze the future discovery of yet unreported macrolactams.

[1] Chemical Biology of Microbe-Host Interactions, Leibniz institute for Natural Product Research and Infection Biology – Hans-Knöll-Institute (HKI), Beutenbergstraße 11a, 07745 Jena, Germany. [2] Anti-Infectives from Microbiota, Helmholtz-Institut für Pharmazeutische Forschung Saarland (HIPS), Campus E8.1, 66123 Saarbrücken, Germany. [3] College of Pharmacy, Yonsei Institute of Pharmaceutical Sciences, Yonsei University, Songdogwahak-ro, Incheon 12983, Republic of Korea. [4] Globe Institute, Section for Molecular Ecology and Evolution, University of Copenhagen, 1350 Copenhagen K, Denmark. [5] Natural History Museum of Denmark – Research and Collections, University of Copenhagen, 2100 Copenhagen East, Denmark. [6] Section for Bioinformatics and Genetics, Swedish Museum of Natural History, 114 18 Stockholm, Sweden. [7] Section for Ecology and Evolution, University of Copenhagen, 2100 Copenhagen East, Denmark. [8] School of Pharmacy, Sungkyunkwan University, Suwon 16419, Republic of Korea. [9] Saarland University, 66123 Saarbrücken, Germany. ✉email: khkim83@skku.edu; Christine.beemelmanns@helmholtz-hips.de

Polyketide-derived macrolactams are a diverse and complex family of natural products with exceptional pharmacological properties, including antibiotic, antifungal, anticancer, and immunosuppressant activities[1]. Macrolactams have been predominately isolated from Actinobacteria of various ecological habitats and are characterized by the presence of an alkyl/alkenyl or aryl-substituted β-amino acid starter unit (Fig. 1) and a polyenylcarboxylic acid (polyketide), connected by an amide bond to form a macrolactam ring of varying size (Figs. S1, S2)[2, 3]. In addition, spontaneous intramolecular cyclization reactions and post-assembly modifications via accessory enzymes largely contribute to the structural diversity of macrolactams[2, 4]. In general, macrolactams can be grouped according to the different types of starter units into five structural categories. Macrolactams that contain the L-glutamate-derived 3-amino-2-methylpropionate (3-Amp) motif include well-studied compounds such as the 20-membered vicenistatin[5], the 22-membered ciromicins A-B[6], as well as 26-membered macrolactams such as sceliphrolactam[7] and bombyxamycins A-B[8]. On the other hand, derivatives carrying a 3-aminobutyrate (3-Aba)-motif include the 24-membered incednine[9], and the 26-membered lobosamides[10], among many other examples (Figs. S1, S2)[2]. Derivatives containing a β-phenylalanine (β-Phe) as an characteristic feature include hitachimycin (stubomycin)[11] and viridenomycin[12], while fluvirucins[13, 14] possess a β-alanine (β-Ala) moiety. Furthermore, macrolactams carrying 3-amino fatty acids (3-Afa)[15–19] have been identified, such as the 20-membered heronamides[20, 21], the 22-membered cremimycin[22], and the largest yet known polyene macrolactam, 34-membered sagamilactam[23]. Based on an ecology-driven approach, we previously reported the isolation of the 20-membered 3-Amp-containing macrolactams termed macrotermycin A–D from a defensive bacterial symbiont, the actinobacterium *Amycolatopsis* sp. M39[24]. Short-read genome sequencing of the producer allowed us to identify a macrolactam-encoding gene cluster region (*mte*) and propose a first, but incomplete biosynthetic pathway (*mte*). In frame of these studies, we realized that despite intensive research efforts, the genetic diversity and regulation of macrolactam encoding gene clusters, and with that the general abundance of macrolactam producers in the bacterial world, has only sparsely been described. Thus, we followed up on our and other groups prior work and evaluated the diversity and organization of macrolactam-encoding biosynthetic gene clusters (BGC) across the bacterial kingdom. Findings of this study uncovered that macrolactam-encoding BGCs are more widespread than originally anticipated and enabled us to revise the structure and composition of the *mte* pathway in *Amycolatopsis* sp. M39. To complement our in silico studies, we employed a comparative metabolomics approach, which enabled the isolation of yet unreported macrotermycin derivatives, as well as the predicted macrolactam ciromicin A from two actinobacterial producers. Additionally, we proved the ciromicin biosynthetic pathway by heterologous production.

## Results and discussion

**Mining of bacterial genomes for macrolactam-encoding BGCs.**
A comprehensive search revealed about 94 literature-reported macrolactam structures of microbial origin by early 2023 (Supplementary Data file 1). Most derivatives belonged to 3-Amp-(32/34%) and 3-Aba-containing macrolactams (22/24%), followed by those derivatives that contain 3-Afa (22/23%), β-Ala

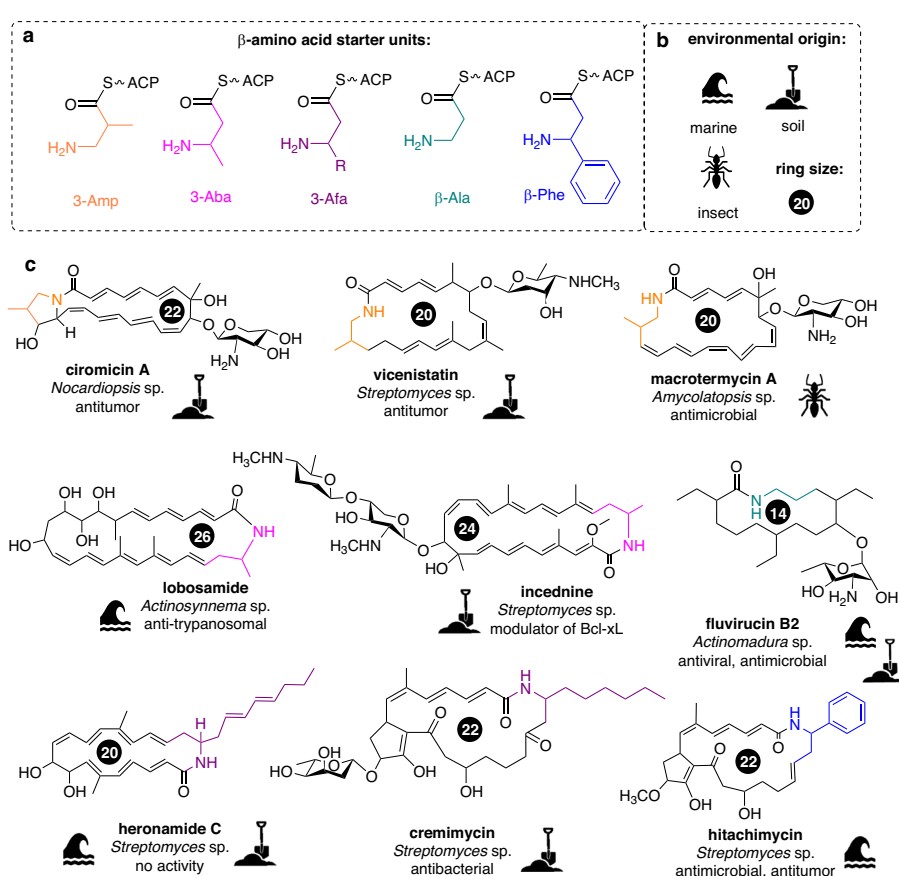

**Fig. 1 Planar chemical structures of β-amino acid starter units and macrolactams.** Planar chemical structures of (**a**) ß-amino acid starter units; (**b**) icons representing the habitat from which the producing organism was isolated, and (**c**) chemical core structures and names of reported macrolactams, producer organism, and major biological activities. The stereochemical assignment of the macrolactam core structure was excluded to enhance clarity.

(16/17%) or β-Phe as starter unit (2/2%) (Fig. S3)[25]. We noted that the majority of the reported producer strains were members of the Streptomycetaceae (34 of total 54 strains, 62%), followed by Pseudonocardiaceae and Micromonosporaceae (6/11% each). A smaller number of producer strains belonged to Thermo-monosporaceae (4/7%), Streptosporangiaceae (1/2%) and other bacterial families (3/6%) (Fig. S4).

We then assessed the distribution and relative abundance of different macrolactam biosynthetic gene cluster types across the bacterial kingdom (Supplementary Data file 2, Table S1)[26], with the assumption that the macrolactam biosynthesis can be broadly categorized into four different phases (Fig. 2). Typically, the first phase involves the biosynthesis of the essential β-amino acid (the starter unit) through a series of enzyme-mediated transformations[27]. In the second phase, the β-amino acid is activated by being loaded onto a discrete acyl carrier protein (ACP) domain by a β-amino acid specific adenylation (A) domain. Afterwards the N-terminus is protected and undergoes further modifications before being loaded onto the starter ACP of a modular type I polyketide synthase (PKS) by a stand-alone acyltransferase (AT) domain[3]. The third phase involves elongation of the polyketide chain using extender units defined by each single module of the PKS. The chain is then cyclized by a thioesterase (TE) domain located within the final PKS module to form the macrocyclic lactam core structure. In the fourth and final phase, the core macrolactam undergoes further modifications by tailoring enzymes, such as glycosyltransferases (GT), methyltransferases (MT), and cytochrome P450 (Ox) enzymes. Due to their polyenic nature, macrolactams have been observed, although not exclusively, to undergo intramolecular cyclization reactions upon abiotic and spontaneous stimuli, leading to further structural diversification of the compound class[4].

Based on these biosynthetic considerations, we selected several conserved biosynthetic enzymes, which are involved in the different phases of precursor and macrolactam biosynthesis (Fig. 3a), for homology searches against the NCBI database using cblaster v1.3.11 (Supplementary Data file 2). As query sequences, we included enzymes involved in amino acid incorporation that are universally present and unique for all macrolactam BGCs, enzymes involved in β-amino acid biosynthesis to distinguish between different BGC types, and one PKS (IdnP1) to account for a type I PKS assembly line encoded in proximity to the starter unit biosynthesis. Due to the highly varying length and module composition of the PKS enzymes in macrolactam BGCs, accurate prediction of PKS homologs was challenging, and hits with ambiguous cluster architecture were curated manually.

Using this query input, we identified 435 genomes, all of which belonged to Actinobacteria and encoded a macrolactam-related biosynthetic gene cluster. The majority of the BGCs (295/68%) was detected in the Streptomycetaceae, followed by Pseudono-cardiaceae (52/12%), Micromonosporaceae (41/9%), Streptospor-angiaceae (23/5%), Thermomonosporaceae (13/3%), and others (11/3%) (Fig. 3b). We also uncovered that 3-Amp and 3-Aba-type BGCs were more broadly distributed, whereas 3-Afa, β-Phe and β-Ala clusters were found only in a few strains (Fig. 3c); a finding that correlated to the abundance and origin of isolated macrolactam natural products (Figs. S4, S5). Here, we acknowl-edge that the relative overrepresentation of BGCs in certain families might also relate to the higher abundance of genomes of e.g., Streptomycetaceae in public databases (Fig. S6). Intriguingly, the distribution of the five macrolactam BGC types varied extensively across bacterial genera (Fig. S7). While all five types were identified in *Streptomyces*, only four were found in *Micromonospora*, three in *Actinomadura*, two in *Actinoplanes*

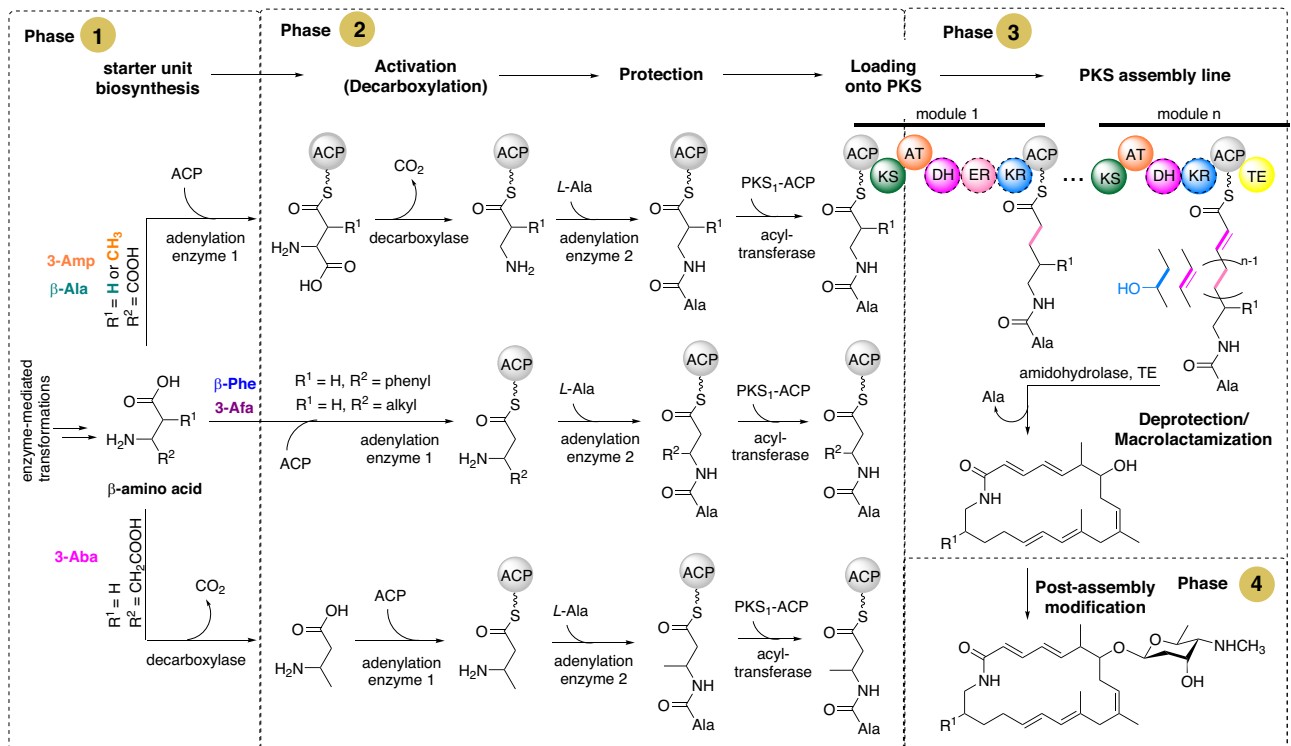

**Fig. 2 The four phases of macrolactam biosynthesis exemplified by vicenistatin biosynthesis.** In the first phase, enzyme-mediated transformations lead to the formation of the β-amino acid starter unit. The second step is common for all macrolactam BGCs and includes activation, protection and loading of the starter unit onto the first PKS. In phase 3, the polyketide chain is elongated by a PKS assembly line following deprotection and macrolactamization. Post-assembly modifications are introduced via accessory enzymes and spontaneous intramolecular cyclization may occur (phase 4). The stereochemical assignment of the macrolactam core structure was excluded to enhance clarity.

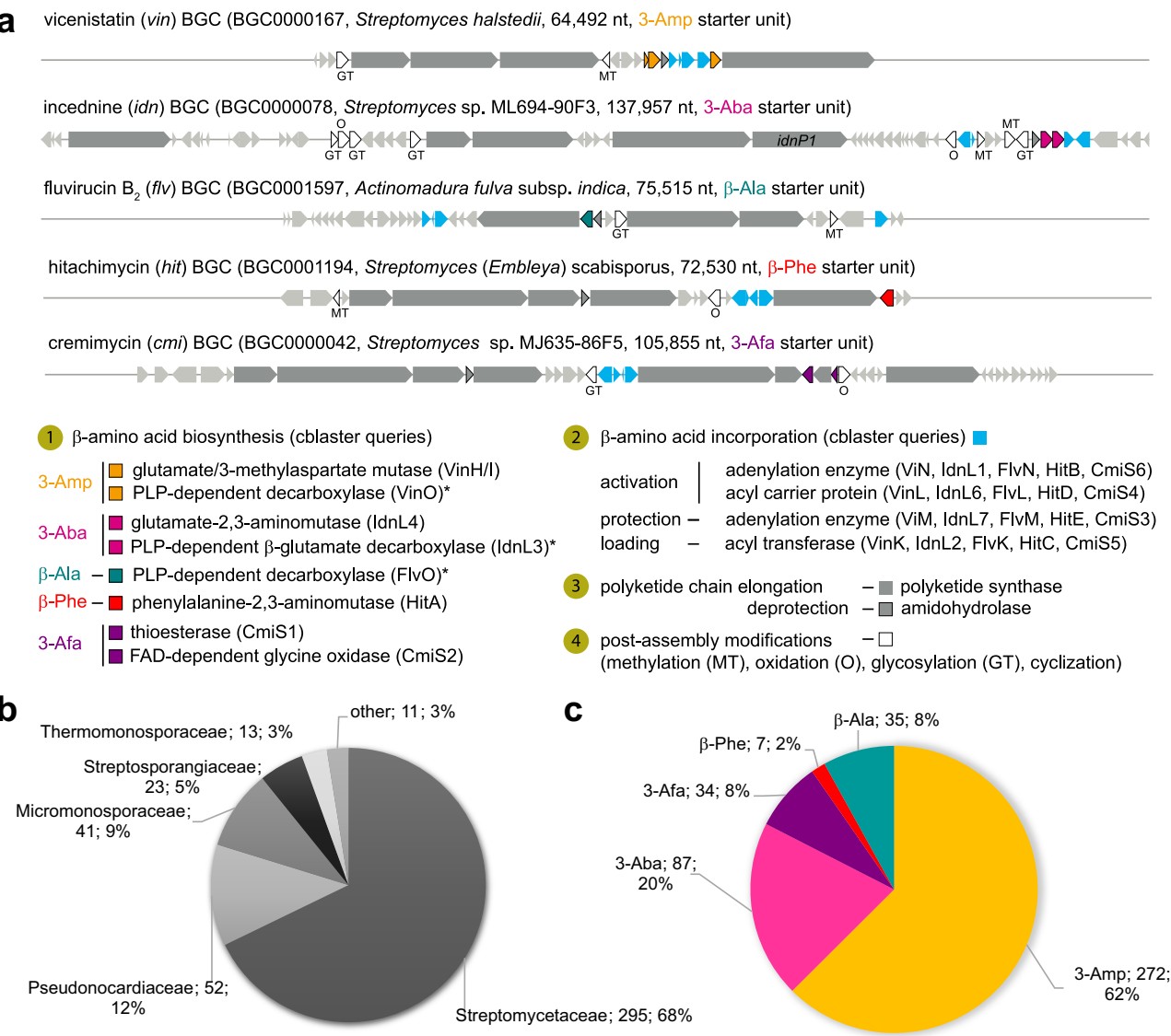

**Fig. 3 Organization and distribution of macrolactam and β-amino acid starter unit biosynthesis encoding biosynthetic gene clusters. a**) A macrolactam-specific cblaster homology search (tblastn) was designed, which included dedicated enzymes involved in β-amino acid biosynthesis (I) and incorporation (II, light blue), and one PKS (IdnP1) to account for the presence of a type I PKS assembly line (gray). **b**) Abundance (absolute; relative) of macrolactam BGCs among bacterial families based on a cblaster analysis against the NCBI database. **c**) Classification of predicted macrolactam clusters by BGC type (according to starter unit) and comparison of their abundance (absolute; relative). Abbreviations: A domain – adenylation domain, ACP – acyl carrier protein, AT domain – acyltransferase domain, GT – glycosyltransferase, MT – methyltransferase, Ox – cytochrome P450 monooxygenase.

and one in *Amycolatopsis*. 3-Amp-type BGCs were found in all genera, while 3-Aba was the dominant starter unit in *Micromonospora*, 3-Afa-types were almost exclusively found in *Streptomyces* and *Actinomadura* and β-Phe BGCs were only identified in *Streptomyces* and *Micromonospora*.

**Revised organization of the *mte* BGC.** Based on these findings, we then reanalyzed the originally described *mte* cluster sequence and found that the sequence encoded less PKS genes than required for a canonical biosynthetic assembly line of the 20-membered macrotermycins, and additionally was lacking the genes related to xylosamine (aminosugar) biosynthesis (Fig. S8)[24]. To address these inconsistencies, we resequenced the strain first with short-read sequencing, which resulted in two differently fragmented *mte* clusters, and then with long-read sequencing that afforded a complete cluster sequence (Fig. S9).

A genome-to-genome distance analysis[28] of the WGS uncovered that strain M39 is very closely related to *Amycolatopsis rubida* DSM44637 (dDDH (d₄): 93.2%) and could be regarded as an isolate of the same species. *A. rubida* DSM44637 encodes a nearly identical macrolactam BGC predicted to generate a 20-membered macrolactam. Sequence alignment of all three obtained genome assemblies with the *A. rubida* genome allowed locating the missing biosynthetic genes in the previously published *mte* BGC (Fig. S9) yielding a complete *mte* BGC region with a predicted PKS domain architecture that matched the formation of the 20-membered macrotermycins (Figs. S9–S12, Table S2). Overall, six PKS genes with ten modules were annotated, of which modules M9 and M10 (MteP3) are proposed to be non-functional due to the lack of crucial domains (Figs. 4, 5). We then compared the organization of the *mte* BGC encoded in the genome of *Amycolatopsis* sp. M39 with the other identified

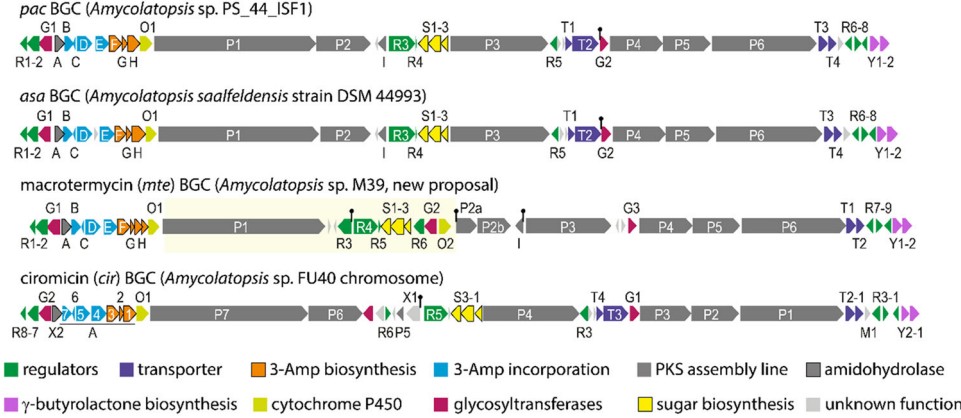

**Fig. 4 Representative comparative cluster gene map of macrolactam BGCs encoding the 3-Amp starter unit biosynthesis in *Amycolatopsis*.** Comparison of the *pac*, *asa*, *cir,* and revised *mte* BGC highlight the different positioning and module arrangement of genes encoding for the PKS and β-amino acid biosynthesis, as well as the varying locations of accessory and regulatory genes.

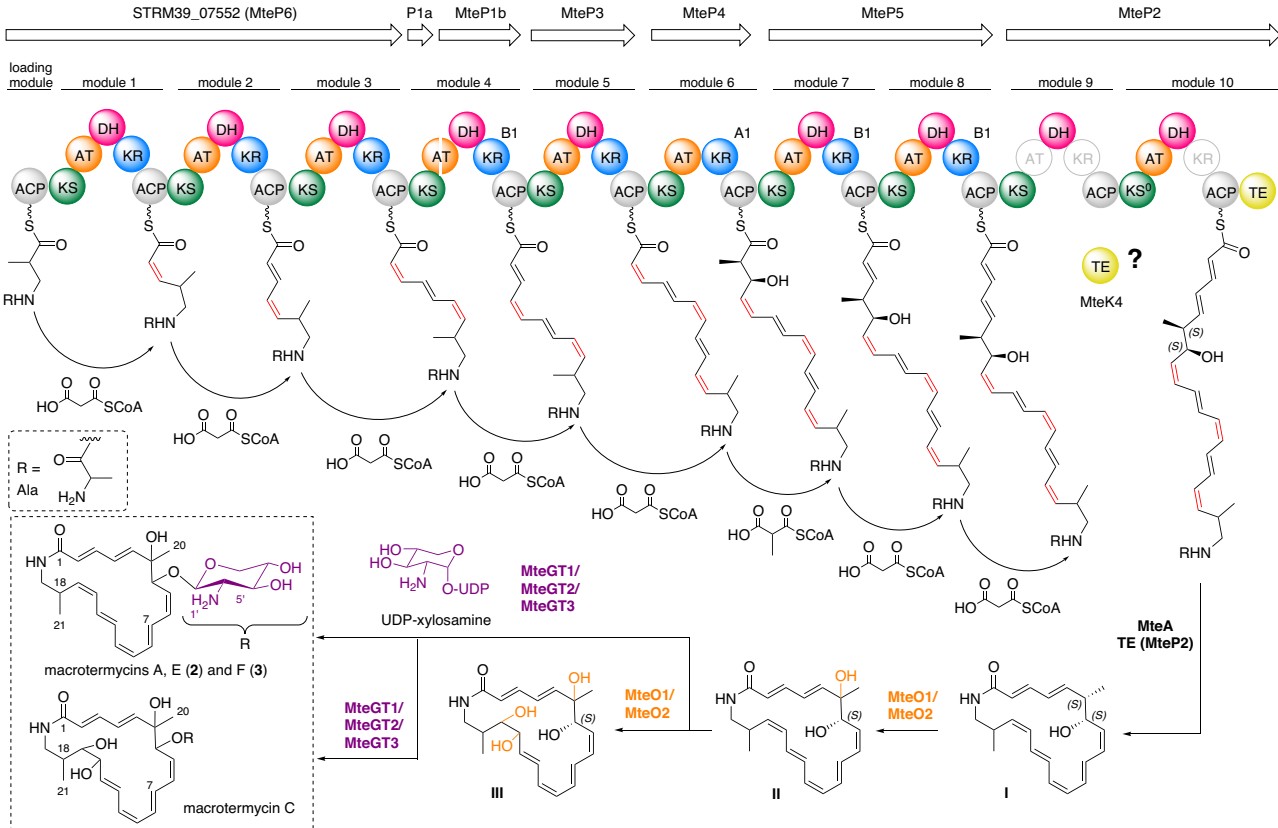

**Fig. 5 Revised proposal for macrotermycin biosynthesis in *Amycolatopsis* sp. M39.** Proposed module organization, biosynthetic intermediates **I–III** and chemical structures of isolated macrotermycins (derivatives A, C, E and F in dashed brackets, for additional details, see Fig 8. (*vide infra*) and Fig. S12). The stereochemical assignment of the macrolactam core structure was partially excluded for clarity.

*Amycolatopsis*-derived BGCs (28) that encode the biosynthesis of the 3-Amp starter unit (Figs. 4, S8) to predict the structural space of Amp-containing macrolactams[29]. All 28 detected BGCs shared a similar architecture, containing genes responsible for 3-Amp biosynthesis (phase 1), incorporation (phase 2), polyketide elongation (phase 3) and modification (phase 4). Most clusters contained six PKS genes encoding for a total number of ten modules with only one exception for *Amycolatopsis xylanica*, which encoded a PKS assembly line with eleven PKS modules resembling the domain architecture of the bombyxamycin (*bom*) BGC[8]. Across the BGCs, the genes for PKS1/2 and PKS4-6

showed identical module composition, while modules of PKS3 lacked certain domains, which could be the cause of varying ring sizes of the macrolactam products. We also observed differences in the number of tailoring enzymes responsible for oxidation (CYPs) or glycosylation (GTs). While many BGCs contained only one CYP and one GT homolog, some clusters encoded for two GT homologs, and others encoded none (Fig. S8). Interestingly, the revised *mte* gene cluster region also contained three glycosyltransferase and two cytochrome P450 encoding genes similar to the macrolactam cluster of *A. rubida*. Alignment of the glycosyltransferase sequences (Fig. S11) revealed that MteG1 and

G2 (420/417 aa) share homologies with AurS5 (55% and 59% identity, respectively), harboring the necessary residues for acceptor and donor binding. In contrast, MteG3 (298 aa) showed similarities to the truncated AurS4 (47% identity), suggesting that MteG3 might act in combination with MteG1 or G2 as exemplified in auroramycin biosynthesis[30]. We also noted the presence of a γ-butyrolactone (γ-BL) biosynthesis subcluster downstream of each macrolactam BGC[31], which may be responsible for the regulation of macrolactam biosynthesis in response to environmental stressors[32].

**Deduction of two putative ciromicin producers.** The comparative study also revealed that our recently discovered bird-associated *Amycolatopsis* sp. PS_44_ISF1 (closest related type strain: *A. saalfeldensis* DSM 44993[33]) carried a macrolactam biosynthetic gene cluster (*pac*), whose arrangement showed high similarities to the *asa* gene cluster of *A. saalfeldensis* DSM 44993 and the ciromicin (*cir*)[6] cluster from *Amycolatopsis* sp. FU40, and partial similarities to the *mte* cluster[24] from *Amycolatopsis* sp. M39 (Fig. 4, Tables S2, S4).

Thus, we pursued a detailed analysis of the predicted protein sequences of the *pac* and *asa* gene cluster to deduce the core structure of the yet unidentified encoded macrolactam. Both gene cluster regions (*pac, asa*) encode for 39–40 open reading frames (*pac*R1-Y2, *asa*R1-Y2 with identical gene arrangement), spanning at least 86.2–86.4 kb (in *asa* BGC an additional gene of unknown function is present between *asa*D and E) (Fig. 6, Table S2). Both clusters contain six PKS genes (P1–6), three β-amino acid biosynthetic genes (F-H), four genes involved in amino acid incorporation (B-E), one cytochrome P450 monooxygenase gene (O1), three xylosamine biosynthetic genes (S1–3), two glycosyltransferase-encoding genes (G1–2), four transporter (T1–4), eight regulatory genes (R1-8), and several other accessory genes. A set of seven genes (*pac*B-H) is predicted to encode for the biosynthesis, activation, and loading of the 3-amino-2-methylaspartate (3-Amp) precursor. In short, the glutamate mutase subunits PacG and PacH are likely responsible for the conversion of L-glutamate to 3-methylaspartate, which should be recognized by adenylation enzyme PacE and transferred onto the standalone ACP PacC. After decarboxylation by decarboxylase PacF to yield 3-Amp, another adenylation enzyme, PacD, transfers L-alanine as a protecting group onto the resulting terminal amino group to prevent internal cyclization during the following PKS assembly line[34]. Subsequently, the *trans*-acting acyltransferase PacB is proposed to transfer the protected precursor onto the first ACP of the type I modular polyketide

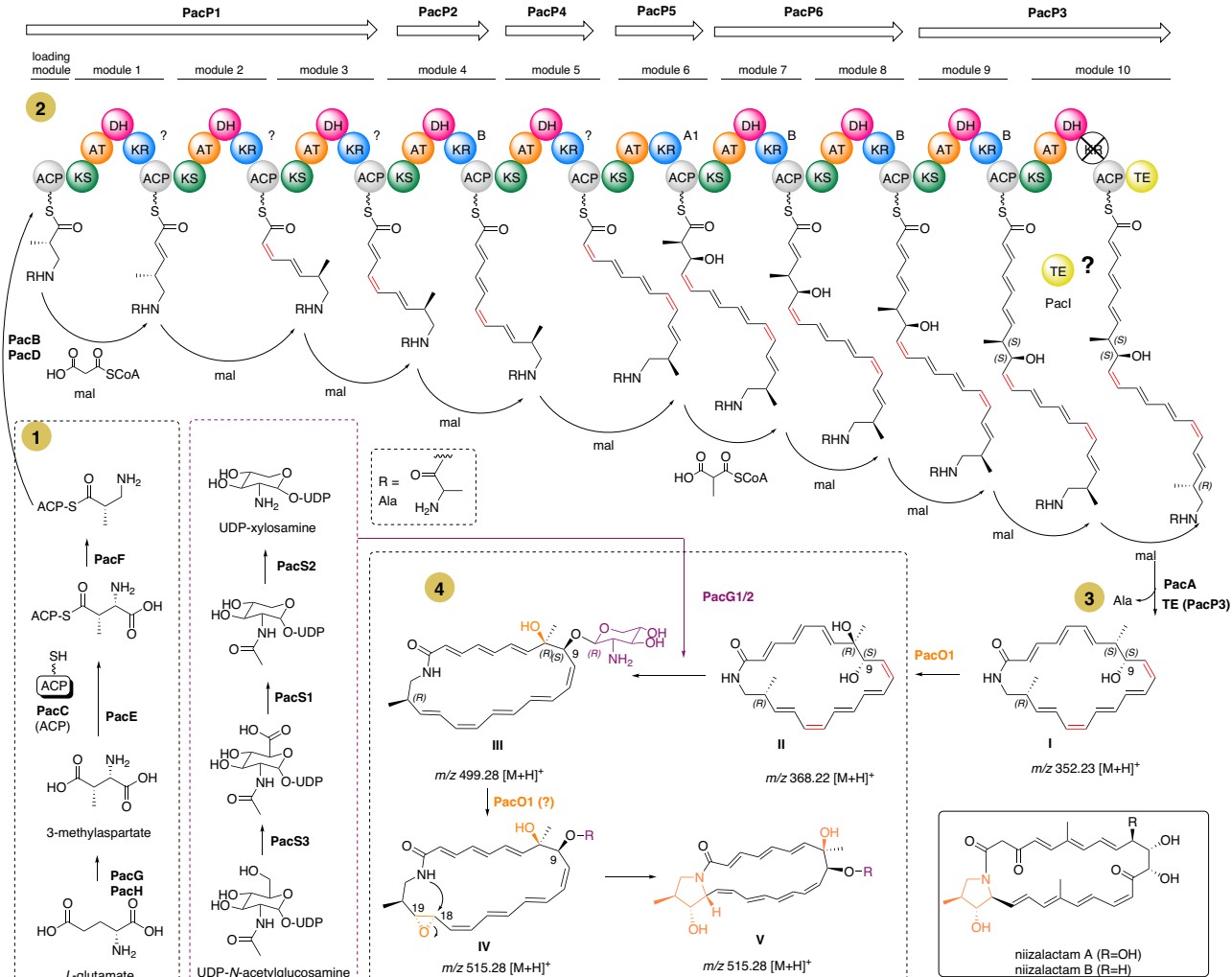

**Fig. 6 Proposed biosynthetic pathway of ciromicin A based on the arrangement of the *pac* gene cluster.** Depicted are the chemical structures of putative biosynthetic intermediates of the precursor biosynthesis (phase 1), intermediates in the PKS assembly line (phase 2) as deduced from the *pac* (*Amycolatopsis* sp. PS_44_ISF1) and *asa* (*A. saalfeldensis*) gene clusters, proposed structures of macrolactam intermediates **I–IV** (phase 4), the structurally characterized and isolated ciromicin A, and the structurally related niizalactams.

synthase assembly line (PKS, *pac*P1-6). Similar to the *cir* cluster[6], the last module 10 of PacP3 lacks a ketoreductase (KR) and harbors an incomplete catalytic triad within the dehydratase domain. This mutation in PacP3 differs from those within the *mte* cluster where both, modules 9 and 10, are proposed to be non-functional. As mutations and variations within the structure of PKS3 are widespread in *Amycolatopsis* macrolactam BGCs, corresponding variations of the macrolactam ring sizes are regularly observed. Lastly, the amidohydrolase PacA is supposed to remove the alanine protecting group, while the terminal thioesterase (TE) domain in PacP3 or the standalone TE PacI catalyzes the lactam formation resulting in intermediate **I**. Assuming a canonical polyketide assembly line, we predicted a 22-membered macrolactam scaffold (intermediate **I**) as aglycon product of this BGC, similar to ciromicin A biosynthesis.

The annotated cytochrome P450 enzyme PacO1 showed high similarity to CirO1 (86% identity), which has been proposed to catalyze the oxidation of the methylated C-8 in ciromicin, hence producing the proposed intermediate **II** (phase 4)[7]. Glycosyl-transferase PacG1 showed high similarity to the putative ciromicin homolog CirG2 (86.4% compared to 24.3% identity for PacG2) and AurS5 (59%)[30] while GT PacG2 was more related to its truncated partner AurS4 (42%).

Due to the high similarity of PacS1–PacS3 with xylosamine-related biosynthetic enzymes within the auroramycin[30] and ciromicin[6] pathways, we deduced xylosamine to be the most likely product of PacS1-PacS3 and to serve as substrate for the encoded glycosyltransferases. Here, we propose that both glycosyltransferases, PacG1 and PacG2 are required for the glycosylation event of position C-9 (intermediate **III**) using xylosamine as reported for the auroramycin biosynthesis. Similar to the ciromicin biosynthesis[6], a second oxidation step (most likely an epoxidation) of the C-18/C-19 double bond (intermediate **IV**), followed by a nucleophilic attack of the amino group at C-18, may lead to the formation of a pyrrolidinol ring likely leading to the ciromicin core structure. A similar mechanism was also proposed for other macrolactams, such as heronamides[35], dracolactams[36], and the depicted niizalactams[37].

We also noted a diverse set of regulatory genes (*pac*R1-8) in both pathways, such as a homolog of the LuxR family transcriptional regulator *aur*R1 (*pac*R3, identity 42%), suggesting a similar regulation of macrolactam biosynthesis in *A. saalfeldensis* and *Amycolatopsis* sp. PS_44_ISF1. Furthermore, homologous genes corresponding to the γ-BL regulation system[32] are close to the BGC (*pac*R5-9, *pac*Y1-2). A third level of regulation may be coordinated by the two-component system of the proposed sensor histidine kinase and its partner response regulator (*pac*R1/2), which also functions in sensing environmental conditions to regulate gene expression in bacteria[38].

**Isolation of predicted macrolactam scaffolds.** In the next step, we evaluated if *Amycolatopsis* sp. PS_44_ISF1 (*pac* BGC) and *A. saalfeldensis* (*asa* BGC) exhibited similar antifungal activity and macrolactam production as *Amycolatopsis* sp. M39 (*mte* BGC) by using bacterial-fungal co-cultivation assays with the ascomycetous fungus *Pseudoxylaria* sp. X802 (Figs. 7a, S13, Tables S3 and S4)[39]. Although isolated from three different geographic locations and habitats, all three strains exhibited comparable antifungal activity towards X802. We then performed a semi-targeted analysis of methanolic extracts of plate cultures of *Amycolatopsis* sp. PS_44_ISF1, *A. saalfeldensis*[40], and *Amycolatopsis* sp. M39 using high-resolution ultra-high performance liquid chromatography-electrospray ionization mass spectrometry (HR-UHPLC-ESI-MS). The acquired MS² data sets were analyzed using the Global Natural Products Social Molecular Networking (GNPS)

platform[41] and results were visualized by Cytoscape (Figs. 7b, S14)[42].

Comparative GNPS-based analysis revealed that the *Amycolatopsis* strains, although closely related and sharing highly similar gene clusters, showed a variety of distinct, and sometimes unidentified metabolite clusters. As expected, clusters containing molecular ion features of the known thiopeptide saalfelduracin A were detectable in *A. saalfeldensis* (Fig. 7d, TIC green)[40], and *Pseudoxylaria*-derived cytochalasins[43] appeared as most dominant features in extracts of the co-culture interaction zones with *A. saalfeldensis*[40] (Fig. 7d, TIC orange). We were also able to identify shared metabolites, predominantly common primary metabolites (membrane lipids, sugars), as well as one cluster related to macrolactams, with nodes ($m/z = 515.276$, $m/z = 499.281$, $\Delta m/z = 16$) present in extracts from *A. saalfeldensis* and strain PS_44_ISF1, and nodes assigned to macrotermycins ($m/z$ 489.258 (macrotermycin C, mteC), $m/z = 473.266$ (macrotermycin A, mteA), $\Delta m/z = 16$, Fig. 7b) in extracts from strain M39. As the macrotermycin-like molecular ion features ($m/z = 515.276$ and 499.281) showed a mass difference of $\Delta m/z = 26$ to macrotermycin A and C, we hypothesized that the detected molecular ion features in *A. saalfeldensis* and *Amycolatopsis* sp. PS_44_ISF1 should contain an additional double bond ($C_2H_2$) and thus likely match the predicted 22-membered macrolactam scaffolds of ciromicin A (**V**) and its precursor **III** (Fig. 6). We then pursued the cultivation only of *Amycolatopsis* sp. PS_44_ISF1 (120 ISP2 agar plates, 150 x 20 mm) as growth studies indicated sufficient macrolactam production for structural characterization (Fig. 7c, d). After seven days of growth, plates were extracted with methanol and compounds were purified by MS- and UV-guided reverse-phase HPLC (Supplementary Note 1, Fig. S15). While we were not able to isolate sufficient amounts of the light-sensitive metabolite **III** ($m/z$ 499.281) for NMR-based structure analysis, we succeeded in the isolation of the macrolactam with molecular ion feature [M+H]⁺ of 515.2766 (colorless solid, 0.6 mg). The observed strong $\lambda_{max}$ of 290 nm and the predicted sum formula $C_{28}H_{38}N_2O_7$ matched with analytical data of the already reported macrolactam ciromicin A, carrying the predicted xylosamine sugar unit ($m/z$ 132.066). Comparative 1D and 2D NMR and HRMS/MS fragmentation pattern analysis confirmed the relative structure as ciromicin A (Figs. 8a, S16–S24, Table S5)[6].

Moreover, we also realized that the molecular ion feature $m/z$ 473.266 ([M+H]⁺) assigned to macrotermycin A in *Amycolatopsis* sp. M39 related to three different retention-times and distinct UV signals (Fig. 7b) indicating the presence of at least two macrotermycin congeners. Thus, we pursued the MS-guided purification of M39 culture extracts and were able to isolate all three derivatives, which were named macrotermycins E-G (Fig. 8B, Supplementary Note 2). Based on 1D and 2D NMR analyses, macrotermycins E and F ($m/z$ 473.2652 [M+H]⁺) exhibited the same planar structure as macrotermycin A[24]. Based on the observed ROESY correlation at H-15/H-18, we propose that macrotermycin E (**2**) and F (**3**) are C-18 stereoisomers of macrotermycins A (Fig. 8). Moreover, when comparing the 2D ROESY correlations of previously isolated macrotermycin A with those obtained from the herein isolated macrotermycins E (**2**) and F (**3**) (Figs. S25–S53, Tables S6–S11), we observed major inconsistencies in the description of the structural assignment of macrotermycin A (stereoconfiguration of C-7 originally depicted as $R^*$)[24]. Based on the herein available comparative analysis, we propose now that the C-7 stereochemistry of macrotermycin A requires revision to $7S^*$, and that macrotermycins E and F are C-7 stereoisomer of macrotermycin A, thus exhibiting a $7R^*$ stereoconfiguration. To support this assumption, we revisited the module organization of the PKS in the

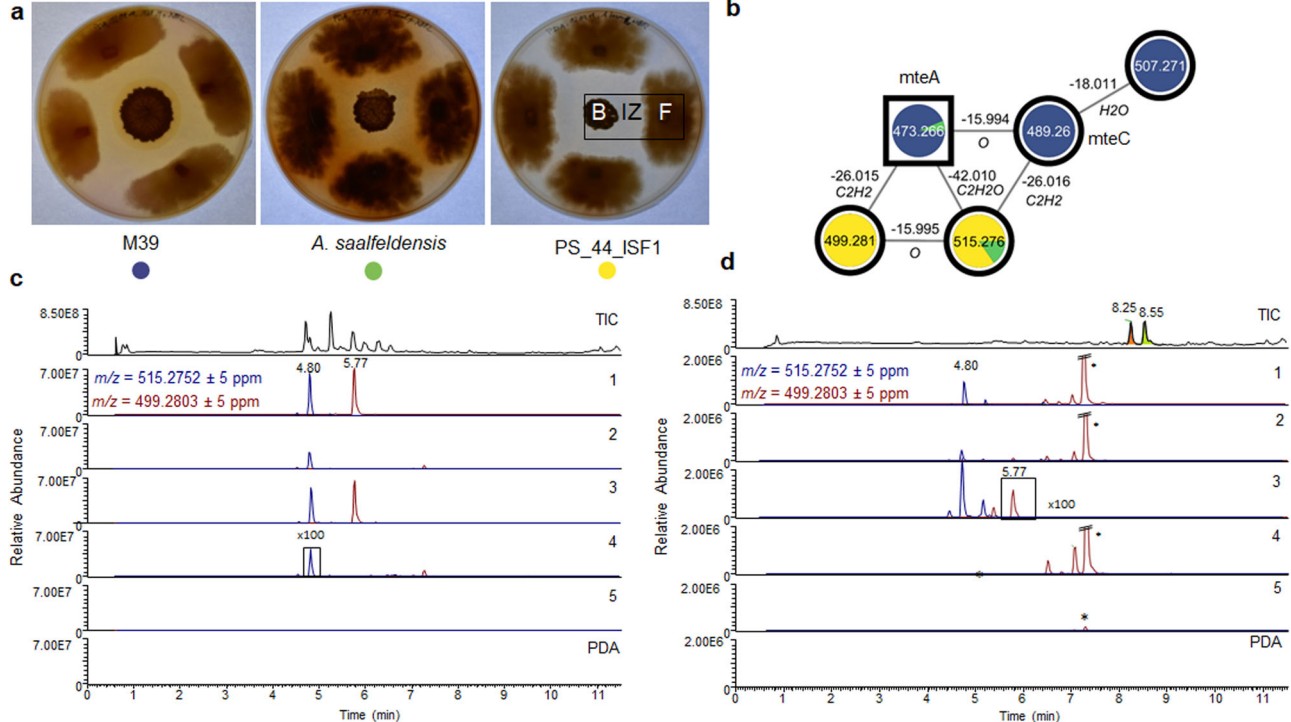

**Fig. 7 Representative co-cultivation assay of *Amycolatopsis* spp. with *Pseudoxylaria* sp. X802. a** Representative co-cultivation plates showing the bacteria *A. saalfeldensis*, *Amycolatopsis* sp. M39 and *Amycolatopsis* sp. PS_44_ISF1 (B), competing fungus (F) and inhibition zone (IZ)[40]. **b** GNPS cluster showing the shared macrolactam cluster detected in *A. saalfeldensis* (green), M39 (blue), and PS_44_ISF1 (yellow). Edge labels depict mass differences between the metabolites and their annotations predicted by GNPS. HRMS chromatograms of methanolic extracts of different regions of the co-cultivation plates and axenic controls of *Amycolatopsis* sp. PS_44_ISF1 (**c**) and *A. saalfeldensis* (**d**). The first lane (TIC) shows the most abundant metabolites in the co-cultivation. The following lanes show EICs of ciromicin A (cir, blue line) and its putative precursor (III, red line) in the culture area B (1), ZI (2), and F (4), as well as in the bacterial (3) and fungal (5) axenic controls and in the medium (PDA). The orange- and green-shaded peaks in the TIC of *A. saalfeldensis* co-cultures represent the fungal cytochalasins and saalfelduracin A, respectively. *Metabolite of fungal origin.

macrotermycin (*mte*) cluster and all ciromicin-related clusters (*pac*, *asa*, *cir*). Signature sequence alignment of all KR domains within modules 1–9 of all four clusters (module 10 lacked a KR domain in all four clusters) clearly showed that all domains share identical KR stereospecificities (Fig. S10)[44]. The KR of module 6 (KR6) was assigned as the only A1-type KR[24], and correlated to a C-7-(*S*)-OH in case of the macrotermycin core structure, and to a C-9-(*S*)-OH in the ciromicin core structure[24,45]. In contrast, KR4 and KR7-9 were assigned as B-type KRs, which would result in an intermediate (*R*)(*R*)-OH configuration at C-2, C-4, C-10 of macrotermycin intermediates (C-2, C-4, C-6 and C-12 of ciromicin A intermediates) before dehydration to (*trans*) double bonds. In contrast, KR1-3 and KR5, could neither be assigned to an A- or B-type KR within all four clusters; however, their hydroxylated biosynthetic intermediates products are likely transformed to double bonds at position C-8, C-12, C-14, C-16 in macrotermycins (C-10, C-14, C-16, and C-18 in ciromicins) by the co-located dehydratase of the respective modules (Fig. S10). Here, it was noted that position C-8 in macrotermycin and position C-10 in ciromicin exhibit an identical double-bound stereochemistry (*cis*), whereas the configurations for all other double bond positions differed (ciromicin A: 14*E*, 16*Z*, 18*E*; macrotermycin A: 12*Z*, 14*E*, 16*Z*). We acknowledge at this stage that studies on the co-linearity between KR domain specificity and double bond geometry still remains debated due to similar numbers of exceptions to this theory as exemplified in the biosynthetic studies on the macrolactam verticilactam[46], and consequently refrain from deducing the double stereochemistry based on sequence alignment only. The isolation of two additional macrotermycin stereoisomers (C-7 and C-18)

indicated that at least two biosynthetic steps within the *mte*-pathway are more promiscuous than originally anticipated: (1) the formation of the β-amino acid 3-Amp and/or its acceptance to the PKS, and (2) the specificity of the ketoreductase KR6 (MteP4). Both findings require future studies to fully elude their selectivity.

Finally, the 1D and 2D NMR datasets of the third isolated congener, macrotermycin G (*m/z* 473.2652 [M+H]$^+$), were deduced to be very similar to those of macrotermycin B, with the most noticeable difference being the replacement of methylene carbon C-2 and oxymethine carbon C-3 by a double bond at C-2/C-3 (Table S11). The double bond geometries of derivative E deduced as 2*E*, 8*Z*, 11*Z*, 14*E*, and 16*Z* were deduced based on the coupling constants observed in the homo *J*-resolved $^1$H NMR spectrum and ROESY correlations. Analysis of coupling constants and H-4/H-13, H-5/H-10, H-5/H-20, H-10/H-20, H-10/H-13, H-7/H-10, H-15/H-18, and H-18/H-21 key ROESY correlations led us to propose the 4*R**, 5*R**, 6*S**, 7*S**, 10*S**, 13*S**, 18*S** relative configuration of macrotermycin G (Fig. 8B).

As the macrolactam scaffold is prone to undergo a light, heat, or acid-induced 2+4 cycloaddition reaction - most likely during work-up procedures —yielding either cycloaddition products of type **I** and after an acid or base-catalyzed 1,4-addition reaction tetracyclic ethers of type **II**, we also conclude at this stage that macrotermycin A and E should serve as cycloaddition precursor for the formation of macrotermycin B and G, respectively (Fig. 8c).

**Heterologous expression of the *asa* BGC.** We then aimed to verify the hypothesized *asa* biosynthetic pathway from *A.*

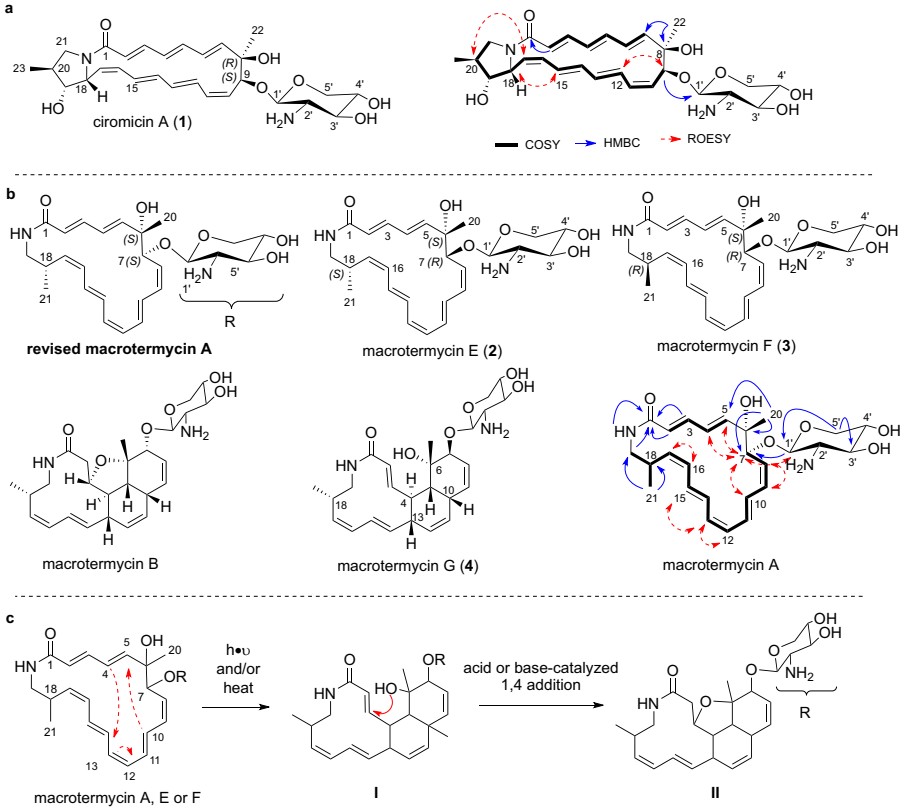

**Fig. 8 Chemical structures of isolated ciromicin A (1) and macrotermycins. a** Deduced chemical structure of ciromicin A based on major 2D NMR correlations. **b** Deduced chemical structure of macrotermycins E-G (**2–4**) and revised chemical structure of macrotermycin A based on comparative 2D NMR analyses. **c** Proposed conversion of macrotermycins of type A after exposure to light and/or heat to intermediate **I** (macrotermycins of type G) via 2+4 cycloaddition reaction, followed by an acid or base-catalyzed 1,4-addition to yield macrotermycins of type B (**II**).

*saalfeldensis* and enable its genetic manipulation by heterologous expression in a *Streptomyces* host. The entire macrolactam cluster region (105.6 kb) including the γ-butyrolactone-related genes and surrounding open reading frames (4 kb left and right of the core cluster) were obtained by in vitro CRISPR-mediated cleavage and transferred into the integrative pDualP vector (Terra Bioforge (Madison, USA)) resulting in the expression vector pAsa (Figs. 9b, S54, Table S12). The vector was chosen as it contained two *Streptomyces*-specific inducible promoters flanking the inserted BGC. The first promotor Potr[47] upstream of the cluster, can be activated by oxytetracycline dihydrate (OXT), while PnitA[48, 49] downstream of the cluster, can be induced in the presence of ε-caprolactam (ε-CL). For heterologous expression of the gene cluster, three *Streptomyces* hosts were selected (*S. lividans* TK24 (1), *S. albus* J1074 (2), and *S. coelicolor* M1146 (3), Fig. 9a) and the integrative expression vector pAsa was introduced via triparental intergeneric conjugational transfer. Exconjugants were confirmed by colony PCR and Sanger sequencing (Fig. 9c, Table S13, and Figs. S55–S56).

To test if macrolactam production is inducible, *S. albus*::pAsa, *S. lividans*::pAsa and *S. coelicolor*::pAsa were cultivated on agar plates using four different growth conditions: (1) ISP2 medium + no induction (basal expression), (2) ISP2 + OXT (induction of promotor Potr), (3) ISP2 + ε-CL (induction of promotor PnitA) and (4) ISP2 + OXT + ε-CL (induction of both promotors). *Streptomyces* wild types (host strain without plasmid) served as control. Agar plates were extracted with EtOAc, and concentrated extracts were analyzed by HRMS/MS and comparative GNPS analysis, using MS/MS data of *A. saalfeldensis* and PS_44_ISF1 as reference (Fig. S57). Unexpectedly, we also noted that the presence of either one of the inducers lowered production titers in *S.*

*albus*::pAsa compared to the basal induction conditions (Fig. 9e, Figs. S58–S60). We also tested different growth conditions (DNPM production medium, ISP2, mannitol-soy flour medium (MS) and vegetative medium (VM), agar and broth, Table S3), but without any notable improvement in yields and production titer (Fig. 9f). Semi-targeted metabolomic analysis uncovered instead that *S. albus* J1074 produced under inducing conditions a variety of intrinsic secondary metabolites, including antimycins, candicidins[50], surugamides[51], and desferrioxamines (Figs. S59–S60)[52]. While we were not able to detect ciromicin-related *m/z* features by targeted or untargeted measures, we found instead HRMS/MS features within the compound cluster that fitted to a proposed biosynthetic intermediate **III** (*m/z* 499.28, see Fig. 6) in *S. albus*::pAsa (Fig. 9d, f). The formation of precursor **III** suggests that the cytochrome P450 enzyme AsaO1/PacO1 proposed to catalyze the oxidation at C-8 and/or C-19 is either dysfunctional when expressed in the host *S. albus*, or not responsible for both oxidation events[6]. Therefore, we cannot exclude that another enzyme located in a different locus of the *Amycolatopsis* genome might account for the second oxidation step, suggesting that the ciromicin precursor **III** does not exhibit a pyrrolidinol moiety (Fig. 6).

## Conclusion

By application of macrolactam-specific query sequences, we uncovered a wide distribution of putative macrolactam producers within the phylum Actinobacteria and were able to differentiate general-specific trends in the abundance of macrolactam BGC types. Our analysis suggested that about 10% of all members of certain actinobacterial genera might harbor a macrolactam-related BGC, with 3-Amp and 3-Aba-type derivatives being most abundant, whereas 3-Afa and β-Phe BGCs seem to be more

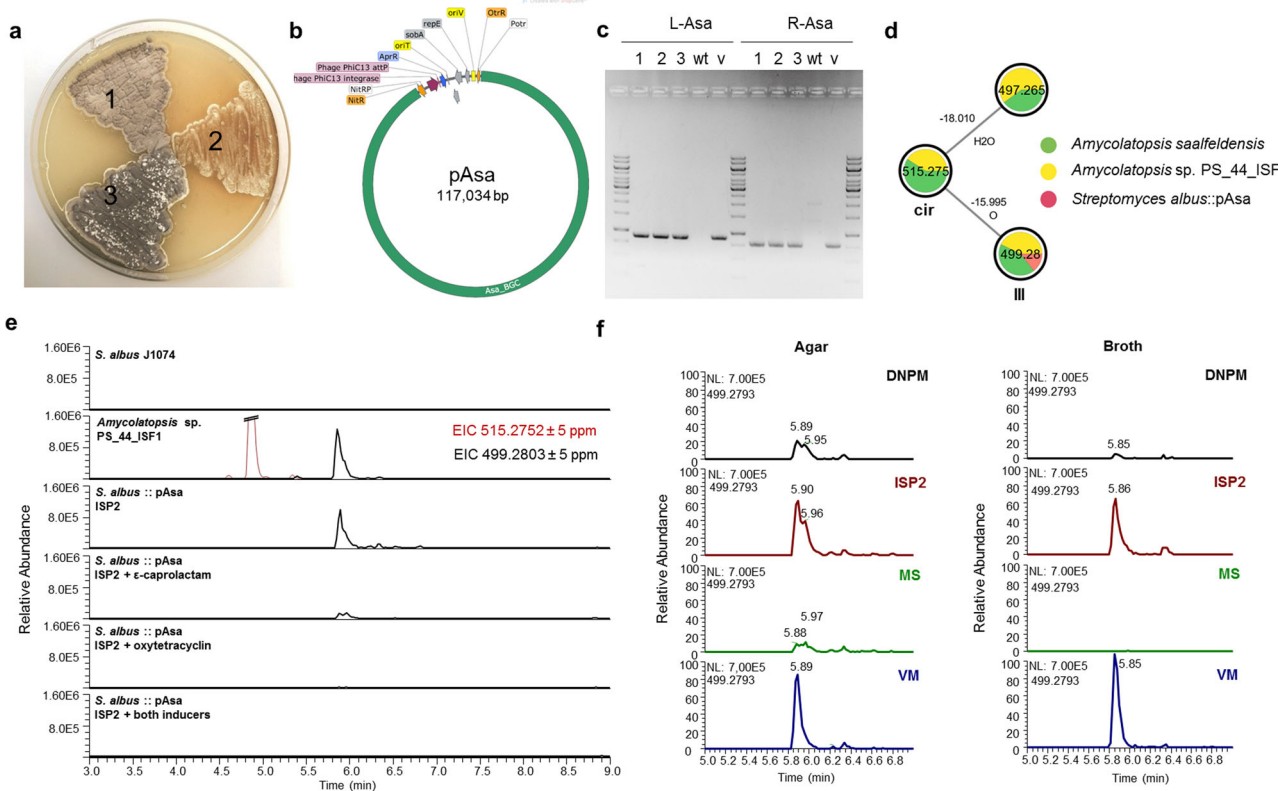

**Fig. 9 Heterologous expression of the *asa* cluster in *Streptomyces* hosts. a** *Streptomyces* hosts: *S. lividans* TK24 (1), *S. albus* J1074 (2) and *S. coelicolor* M1146 (3). **b** Vector map of the inducible expression vector pDualP harboring the *asa* cluster. For more details, see Fig. S54. **c** Verification of pAsa insertion in *S. albus* exconjugants 1–3 using primers L-Asa and R-Asa (w – wild type control, v – vector control). **d** GNPS cluster of macrolactams produced by *Amycolatopsis* spp. and heterologous strain *S. albus*::pAsa. **e** Extracted ion chromatograms of intermediate **III** (Fig. 6, EIC 499.2803 ± 5 ppm) in *S. albus* exconjugants carrying the *asa* BGC cultivated on ISP2 with different inducers. *S. albus* wildtype served as negative control and *Amycolatopsis* sp. PS_44_ISF1 as positive control. **f** Production of ciromicin precursor **III** by *S. albus*::pAsa in four different liquid and agar media.

exclusive for some bacterial genera. In-depth analysis of macrolactam BGCs encoded in *Amycolatopsis* spp. revealed that mutations within certain PKS regions are the major cause of the varying ring sizes of the produced macrolactam. Our comparative analysis also showed that macrolactam BGCs contain three main systems of regulation, including LuxR family regulators, sensor histidine kinase two-component systems and γ−butyrolactone regulatory systems. This suggests a complex regulation of macrolactam biosynthesis, and understanding these mechanisms and the environmental stressors that might activate transcription of macrolactam BGCs will help to discover their full potential. Based on our genomic and metabolomic analyses, we identified two *cir*-like gene cluster homologs in *Amycolatopsis* sp. PS_44_ISF1 (*pac*) and *A. saalfeldensis* (*asa*), respectively, and were able to verify the prediction by isolation of ciromicin A. Furthermore, we were able to pursue the heterologous expression of the *asa* BGC in *S. albus* J1074, which allows for future manipulation of the macrolactam pathway to deduce the biochemistry and promiscuity of single biosynthetic steps. In addition, metabolomic analysis led to the isolation of three to the best of our knowledge unreported macrotermycin derivatives from *Amycolatopsis* strain M39 and comparative NMR analyses, enabled us to determine their relative stereoconfiguration and revise the stereochemistry of macrotermycin A.

## Methods
### General experimental procedures
*Chemicals.* All media and ingredients and chemicals were purchased as follows: Methanol (Th. Geyer, Renningen); water for

analytical and preparative HPLC (Millipore, Germany), formic acid (Carl Roth, Germany); acetonitrile (Th. Geyer, Renningen), DMSO (Carl Roth, Germany), media ingredients (Carl Roth, Germany).

*Strains and cultural conditions.* *Amycolatopsis* spp. were subcultured on MS agar. The bacteria were stored in 25% glycerol at −80 °C. Culture media (Table S3) were autoclaved at 121 °C for 20 min. *E. coli* was cultivated on Luria-Bertani (LB) medium/agar. For selection of transformants and ex-conjugants, media were supplemented with antibiotics (depending on the purpose): 50 µg/mL apramycin (Apr), 25 µg/mL chloramphenicol (Cam), 25 µg/mL kanamycin (Kan), or 25 µg/mL nalidixic acid (NA). If not stated differently, all media components were purchased at Carl Roth, Karlsruhe.

*HR-MS.* High-resolution ultra-high performance liquid chromatography electrospray ionization mass spectrometry (HR-UHPLC-ESIMS) measurements were carried out on a Dionex Ultimate3000 system (Thermo Scientific) combined with a Q-Exactive Plus mass spectrometer (Thermo Scientific) equipped with an electrospray ion (ESI) source and a Luna Omega C18 column (100 × 2.1 mm, particle size 1.6 µm, pore diameter 100 Å). The column oven was set to 40 °C, scan range of MS was set to *m/z* 200 to 2000 with a scan speed of 10,000 u/s and event time of 0.25 s under positive mode. DL temperature was set to 250 °C with an interface temperature of 350 °C and a heat block of 400 °C. The nebulizing gas flow was set to 1.5 L/min and dry gas flow to 15 L/min. A combination of data-dependent MS/MS

analysis and Top10 experiments was applied, and measurements were performed in centroid mode with a resolving power of 17,500 at $m/z$ 200, an isolation window of 1 $m/z$ and stepped normalized collision energy of 20/30/40. For structure elucidation MS/MS spectra for **1** and its congeners were predicted using CFM-ID 4.0[53] and compared to the experimental MS/MS spectra.

*GNPS analysis.* Tandem mass spectrometry molecular networks were created using the GNPS platform (http://gnps.ucsd.edu). Data were first converted to the. mzML format with MS-Convert[54]. The converted files were used to generate an MS/MS molecular network using the GNPS Data Analysis workflow version release 30. The precursor and fragment ion mass tolerance were set to 0.02 Da and to a product ion tolerance of 0.02 Da. Networks were generated using four minimum matched ion fragments, a minimum cluster size of 2 and a cosine score of 0.75–0.8, depending on the experiment. The remaining parameters were kept at default. The library spectra were filtered in the same manner as the input data. All matches kept between network spectra and library spectra were required to have a score above 0.7 and at least six matched peaks. After analysis, data were opened and visualized using Cytoscape 3.8.0 software[42].

*Analytical and preparative HPLC.* The high-performance liquid chromatography (HPLC) was carried out by using an Agilent 1290 UHPLC system (Agilent Technologies, Santa Clara, California, United States), which was outfitted with a 1290 Infinity binary pump. For the semi-preparative HPLC, analytical grade solvents were used by J.T. Baker (Philipsburg, NJ, USA) and Burdick & Jackson (Muskegon, MI, USA).

*NMR.* NMR spectra were collected using an NMR spectrometer outfitted with a Bruker CryoPlatform (DMSO-$d_6$, 700 MHz and 900 MHz) at the Korea Basic Science Institute in Ochang, Korea. MestReNova ver. 12.0.1 was used to process the data for the NMR spectra.

*Detection of macrolactam BGCs in public genomes.* To evaluate the distribution of macrolactam BGCs among bacterial families and to compare the relative abundance of the different macrolactam BGC types, homology searches were conducted using cblaster v1.3.11[26]. Key biosynthetic genes with unique occurrence in macrolactam BGCs from four representative BGCs (vicenistatin, incednine, cremimycin, hitachimycin) were chosen as query sequences, such as 1) the VinN-type adenylation enzyme, which transfers an amino acid starter unit on the stand-alone VinL-type ACP, 2) the VinM-type adenylation enzyme protecting the free amino group by aminoacylation with *L*-alanine, and 3) the VinK-type acyltransferase responsible for the transfer of the dipeptidyl group to the loading-ACP of the first PKS enzyme in the assembly line[3]. To be able to distinguish between the different macrolactam BGC types, sequences of the enzymes responsible for the biosynthesis of the β-amino acid starter unit were included: (1) glutamate mutase (VinH, VinI) and decarboxylase (VinO/FlvO)[5, 13] for BGCs containing 3-amino-2-methylpropionate (3-Amp) and β−alanine starter unit (β-Ala), (2) glutamate-2,3-aminomutase (IdnL4)[9] and decarboxylase (IdnL3) for 3-aminobutyrate starter unit (3-Aba), (3) phenylalanine-2,3-aminomutase (HitA) for BGCs with β-phenylalanine starter unit (β-Phe), (4) thioesterase (CmiS1) and flavin adenine dinucleotide (FAD)-dependent oxidase (CmiS2) responsible for the biosynthesis of 3-amino fatty acid starter units[55]. In addition, one PKS protein sequence (IdnLP1) was added to each set of queries to enable the detection of PKS genes in proximity of the starter biosynthesis. However, it must be noted that due to the

highly varying size and module composition of the PKS enzymes in macrolactam clusters, PKS homologs are not reliably detected (cut off pairwise identity 50%) Due to the size of macrolactam BGCs (~60–120 kb) analysis default parameters were adjusted: The maximum distance between any two hits in a cluster ($-g$, default: 20,000 bp) was set to 80,000 bp accounting for the possibility of long gaps between two hits, e. g. interrupted by long PKS genes. The minimum number of total hits in a cluster (*-u*) was set to N-2 ($N$ = number of query sequences) to discard clusters which lack homologs to more than two query sequences. For β-Ala BGCs this parameter was set to $N − 4$ to detect clusters with the absence of VinH/I homologs. The parameters *-ig* and *-md 70,000* were set to enable the detection of intermediate genes ($−ig$) between and adjacent to the homology hits in one cluster (maximum distance (*-md*): 70,000 bp). These parameters enable the visualization of surrounding genes using the *plot_clusters* argument to check for the presence of neighboring PKS genes which are also essential for macrolactam biosynthesis. After analysis, cluster hits were curated manually by firstly removing all duplicates within one BGC type. Cluster hits with unclear cluster architecture (e.g., on contig edge with no PKS genes in proximity) were re-evaluated manually using antiSMASHV6.0[56] and removed from the analysis if necessary. Then, shared hits between two BGC types were detected to evaluate the specificity of the chosen query sequences for one BGC type and to detect duplicates. Duplicates were uploaded to antiSMASH V6.0[56] and assigned to the appropriate BGC type. Curated results were summarized and sorted by bacterial genera and families.

## Whole-genome sequencing of *Amycolatopsis* sp. M39

*DNA extraction.* Cultures of *Amycolatopsis* sp. M39 (50 ml) were prepared in ISP2 broth and cultivated for 7 days at 30 °C and shaking at 150 rpm. Bacterial pellets were harvested, frozen in liquid nitrogen, and grounded to a fine powder. DNA was extracted using DNeasy Plant Mini Kit (Quiagen) according to manufacturer's instructions and subsequently precipitated using isopropanol. The precipitated DNA was washed with 95% ethanol, dried, and dissolved in water before subjecting it for sequencing.

*Whole genome sequencing.* Whole-genome sequencing was performed using a 150 bp paired-end shotgun Illumina sequencing at KIT, Germany. Additionally, Oxford Nanopore technology (Oxford Nanopore Technologies, Oxford, UK) was employed for long-read sequencing. Library preparation for ONT long-read sequencing was performed using the Ligation Sequencing (SQK-LSK109) and Native Barcoding Expansion 1–12 (EXP-NBD104) kits. The ONT protocol for native barcoding genomic DNA to multiplex 12 samples on one flow cells was used. A 48 h sequencing run was performed with the MinION Mk1B device and a 9.4 flow cell using the "super accuracy" basecalling in MinKNOW (v22.12.7). Basecalling and demultiplexing was automatically executed by the integrated basecaller GUPPY (v6.5.6). As a first step of genome assembly, read ends were trimmed manually using SNIKT (Slice Nucleotides Into Klassifiable Sequences v0.5.0)[57] removing the first 75 and last 60 bases. Afterwards, mild long-read polishing was applied with Filtlong (v0.2.1) removing reads below 1 kbp length[58]. Filtlong was used to remove 10% of low-quality length-filtered reads based on read quality[59]. Subsequently, the polished reads were assembled with Flye (v2.9.2) and rotated with rotated according to the Wick et al. perfect bacterial genome assembly tutorial prior to the second polishing step[60]. Lastly, the long-read genomes were polished using Medaka (-m r941_min_sup_g507, v1.8.0)[61].

*Co-cultivation studies.* Co-cultivations of *Pseudoxylaria* sp. X802 with *Amycolatopsis* spp. were performed[40]. In short, a 50 μL droplet of a 7 day-old 20 mL *A. saalfeldensis*, *Amycolatopsis* sp. PS_44_ISF1 or *Amycolatopsis* sp. M39 pre-culture in ISP2 was placed at the center of a 150 × 20 mm PDA plate and incubated for 7 days at 28 °C. Afterwards four 1 cm² agar pieces of a 2-week-old *Pseudoxylaria* sp. X802 culture were placed around the bacterium with a distance of 3 cm. Plates with only X802, *A. saalfeldensis*, PS_44_ISF1 or M39 served as axenic culture controls. Each co-cultivation and control set-up was prepared in five replicates. Cultures were incubated at room temperature for 3 weeks. Pictures were taken on day 1, 7, 14, and 20 of incubation. After 20 days samples (five agar plugs per sample) were taken from the bacterial colony, the fungal mycelium, and the inhibition zone of the co-cultures. Controls were sampled from axenic bacterial and fungal control plates. Samples (five agar plugs per sample) were transferred into a pre-weighed 250 mL Erlenmeyer flask and extracted twice with 100 mL of MeOH overnight at room temperature. The samples were passed through a filter and the solvent was concentrated under reduced pressure. The crude extract was dissolved in 20% MeOH and loaded on an activated SPE-C18 (100 mg) column, washed with 20% MeOH followed by stepwise elution with MeOH/H$_2$O mixtures (50–100% MeOH). The elution fractions were combined and concentrated in vacuo. Samples were dissolved in MeOH to a concentration of 50 μg/mL and analyzed by HR-UPLC-MS/MS. The methanolic extracts of the *Amycolatopsis* strains in co-cultivation were compared using GNPS (cosine score: 0.75). To identify fungal metabolites in the co-cultivation extracts, methanolic extracts of X802 in co-cultivation and axenic cultures were added. Lastly, PDA was used as control to remove metabolites originating from the medium.

*Isolation of ciromicin A.* For metabolite extraction from agar plates, 500 μL of a 7-day-old pre-cultures of *A. saalfeldensis* and strain PS_44_ISF1 were streaked on 120 ISP2 agar plates (150 × 20 mm). The cultures were sealed with parafilm and incubated at 30 °C for another 7 days. Plates were cut into small cubes (1 × 1 cm) and extracted with 6 L isopropanol in the dark overnight. Afterwards, the solvent was filtered off and evaporated in vacuo. The dried extract was then redissolved in 20% methanol and subjected to solid phase extraction (SPE). To remove media components, a 10 g C$_{18}$ column was activated with 100% methanol (MeOH) (2 column volumes (CV)) and equilibrated with 20% methanol. Afterwards the extract was loaded, washed with 20% methanol, and eluted with 50% and 100% MeOH (2 CV). The latter fractions were concentrated under reduced pressure, resuspended in MeOH, adsorbed on Celite and dried in vacuo. Prepacked C18 Sepak resin (2 g) was then loaded with the Celite-adsorbed extract. The extract was fractionated using an elution gradient of water and MeOH. Ciromicin A eluted in the 40% MeOH fraction, and this fraction was purified using semi-preparative reversed-phase HPLC (ODS-A C18, 5 μm silica gel, 150 × 10.0 mm; YMC, Japan; flow rate of 3 mL/min; gradient solvent system: 0–5 min (13% aqueous acetonitrile), 5–20 min (13–20% aqueous acetonitrile over 15 min)) to obtain the pure ciromicin A which eluted at 16.9 min, yielding 0.6 mg from culture extracts of strain PS_44_ISF1 (See Supplementary Note 1).

*Isolation of macrotermycins.* We performed a preparative scale fermentation of *Amycolatopsis* sp. M39 on solid ISP2 agar for 14 d at 30 °C (see Supplementary Note 1). Agar plates densely covered with mycelium were extracted using a mixture of MeOH and *i*PrOH. The crude extract containing macrotermycins was first purified using an activated pre-packed C18 Sep-Pak cartridge and SPE fractions eluted with 80% and 100% MeOH. Enriched extracts were purified by repetitive purification by semi-preparative reverse-phase HPLC equipped with a phenyl-hexyl column and using variations of a gradient program with mixtures of water (A) and methanol (B); 0–5 min: 40% B; 5–30 min: from 40% to 100% B; 30–40 min: 100% B; 41–50 min: isocratic 40% B; total 50 min, flow rate: 10 mL/min. In total, we were able to isolate 0.8 mg of **2**, 0.7 mg of **3** and 0.9 mg of **4** from 6 L of ISP2 agar (80 plates) (See Supplementary Notes 2 and 3).

**Physical data. Ciromicin A (1):** colorless oil; $[\alpha]^{25}_D = +24.2$ (*c* 0.04, MeOH); IR (neat) $\nu_{max}$ 3340, 2955, 2830, 1665, 1595, 1470, 1380 cm$^{-1}$; UV, $\lambda_{max}$ 290 nm; $^1$H and $^{13}$C NMR data, see Table S5; HR-ESI-MS *m/z* 515.2766 [M + H]$^+$ (calcd for C$_{28}$H$_{38}$N$_2$O$_7$, 515.2763)

**Macrotermycin E (2):** yellowish amorphous powder; $[\alpha]^{25}_D = +57.5$ (*c* 0.08, MeOH); IR (neat) $\nu_{max}$ 3315, 2935, 2819, 1660, 1591, 1530, 1450, 1394, 1150 cm$^{-1}$; UV (MeOH) $\lambda_{max}$ (log ε) 240 (4.0), 260 (4.1), 338 (2.6) nm; $^1$H (DMSO-$d_6$, 600 MHz) and $^{13}$C NMR (DMSO-$d_6$, 150 MHz) data, see Tables S6–S8; $^1$H (CD$_3$OD, 600 MHz) and $^{13}$C NMR (CD$_3$OD, 150 MHz) data, see Tables S6, S7; positive HR-ESIMS *m/z* 473.2652 [M+H]$^+$ (calcd for C$_{26}$H$_{37}$N$_2$O$_6$, 473.2652).

**Macrotermycin F (3):** yellowish amorphous powder; $[\alpha]^{25}_D = +46.6$ (*c* 0.06, MeOH); IR (neat) $\nu_{max}$ 3298, 2951, 2829, 1652, 1592, 1531, 1458, 1390, 1148 cm$^{-1}$; UV (MeOH) $\lambda_{max}$ (log ε) 238 (4.0), 261 (4.1), 333 (3.1) nm; $^1$H (CD$_3$OD, 600 MHz) and $^{13}$C NMR (CD$_3$OD, 150 MHz) data, see Table S9; positive HR-ESIMS *m/z* 473.2655 [M+H]$^+$ (calcd for C$_{26}$H$_{37}$N$_2$O$_6$, 473.2652).

**Macrotermycin G (4):** amorphous powder; $[\alpha]^{25}_D = +11.8$ (*c* 0.01, MeOH); IR (neat) $\nu_{max}$ 3355, 2955, 2835, 1660, 1595, 1530, 1470, 1385, 1150 cm$^{-1}$; UV (MeOH) $\lambda_{max}$ (log ε) 225 (3.8), 280 (2.0) nm; $^1$H (CD$_3$OD, 600 MHz) and $^{13}$C NMR (CD$_3$OD, 150 MHz) data, see Tables S10, S11; positive HR–ESIMS *m/z* 473.2650 [M+H]$^+$ (calcd for C$_{26}$H$_{37}$N$_2$O$_6$, 473.2652).

### Heterologous expression of *asa* BGC from *A. saalfeldensis*

*Construction of the expression vector pAsa.* For heterologous expression of the *asa* BGC from *A. saalfeldensis* an inducible dual-promotor expression vector harboring the complete *asa* BGC was constructed by Terra Bioworks (pAsa). For vector construction, isolated gDNA of *A. saalfeldensis* was cut with Cas9 using guide RNA sequences (Table S12). The Cas9 digestion was monitored by qPCR assays using primers designed for both the left and right cut sites of the cluster and was verified to have greater than 70% efficiency before moving forward with DNA assembly. DNA assembly of the genomic preparation was then performed to linearized pDualP vector containing overlap regions specific to the target fragment. The DNA assembly reaction was transformed to *E. coli* BacOpt2.0 from which we recovered colonies (21 transformants, 62% correct by colony PCR). Colonies were assayed via colony PCR using primers for both the left and right cloning junctions and confirmed by Sanger sequencing. Clones for the construct were restriction digested separately by EcoRI and BamHI and compared to a simulated digest to confirm the insertion of the whole BGC into the vector backbone.

*Conjugational transfer of the expression vector to Streptomyces.* For heterologous expression, three *Streptomyces* hosts were chosen: *S. albus* J1074, *S. coelicolor* M1146 and *S. lividans* TK24. The integrative expression vector pAsa was introduced into *Streptomyces* spp. via intergeneric conjugational transfer with minor modifications[62]. For a triparental conjugation, 10 mL *E. coli* BacOpt2.0::pAsa and the helper strain *E. coli* K12 HB101::pRK2013 in LB medium (supplemented with 50 μg/mL

Apr for pAsa, 50 μg/mL Kan for pRK2013) were grown at 37 °C to an $OD_{600}$ of 0.4–0.5. Cultures were centrifuged at $6000 \times g$ for 15 min, washed twice with ice-cold LB to remove residual antibiotics, and resuspended in fresh 1 mL LB without antibiotics. *Streptomyces* spore suspensions (50 μL, $2 \times 10^8$ spores, in 25% glycerol) were added to 500 μL of $2 \times$ YT medium and heat shocked at 50 °C for 10 min. Equal amounts (v/v) of donor (*E. coli* mixture 1:1) and recipient cells (*Streptomyces* spores) were mixed, and the bacteria were pelleted by centrifugation. The medium was discarded, and the pellets were resuspended in a residual 100 μL of medium. The suspension was streaked on MS agar plates, supplemented with 10 mM $MgCl_2$. After incubation for 16–20 h, the plates were overlaid with 1 mL of water containing 0.5 mg of nalidixic acid (NA) and 1.25 mg of apramycin, and incubation at 30 °C was continued for 5–7 days. Exconjugants were streaked on MS (with 50 μg/mL Apr and 25 μg/mL NA). For verification of vector insertion, a PCR using the primer pairs L-Asa and R-Asa was conducted (Table S13). PCR conditions were as follows: 98 °C/5 min, 30 cycles (98 °C/ 30 s, 61 °C/45 s, 72 °C/30 s), 72 °C/10 min. Reactions were performed in GC buffer supplemented with 5% DMSO. PCR products were then purified using a PCR gel extraction kit and sequenced via Sanger sequencing (Eurofins genomics).

*Metabolic analysis of Streptomyces heterologous hosts.* For analysis of macrolactam production in the heterologous hosts carrying the integrative and inducible expression vector pAsa. This vector is equipped with two Streptomyces-specific, inducible promoters flanking the inserted BGC. Potr[47], upstream of the cluster, can be activated by oxytetracycline dihydrate (OXT) and PnitA[48, 49], downstream of the cluster, can be activated by ε -caprolactam (ε-CL). To test whether macrolactam production can be induced using one or both promotors, Streptomyces were tested in four different conditions: ISP2 + no inducer (basal expression), ISP2 + OXT (final concentration 2.5 μM), ISP2 + ε-CL (final concentration 0.1%) or ISP2 + both inducers simultaneously. First 20 ml pre-cultures of *S albus*::pAsa, *S. coelicolor*::pAsa, and *S. lividans*::pAsa in ISP2 (+ 50 μg/mL Apr, 25 μg/mL NA) were grown at 30 °C and 150 rpm for 7 days. Afterwards, 100 μL of the preculture were inoculated on ISP2 agar (plates $90 \times 15$ mm) with different inducer combinations and incubated at 30 °C for another 7 days in the dark. One plate of each Streptomyces strain and each treatment was cut into small cubes ($1 \times 1$ cm) and extracted with 30 mL EtOAc overnight. The solvents were evaporated under reduced pressure and the dried samples were dissolved in MeOH to a concentration of 50 μg/mL and analyzed by HR-UPLC-MS/MS. To compare macrolactam production with producer strains MS/MS data was also used for GNPS analysis.

*Media study to optimize macrolactam production.* To optimize macrolactam production in *S. albus*::pAsa, production was tested on four different media (broth and agar): DNPM, ISP2, MS, VM. For agar cultivation, two agar plates of each medium ($90 \times 15$ mm, 50 mL) were inoculated with 100 μL of a 7 day-old pre-culture of *S. albus*::pAsa in ISP2. For broth cultures, 50 ml of liquid medium were inoculated with 500 μL of the same pre-culture. After another 7 days of cultivation, agar plates were cut into small cubes ($1 \times 1$ cm) and extracted with 50 mL MeOH overnight. Liquid cultures were filtered, and the medium was directly subjected to an SPE column (200 mg). On the next day, agar extracts were filtered, and the solvent was evaporated under reduced pressure. The raw extract was redissolved in 20% MeOH and subjected to an activated and equilibrated 1 g $C_{18}$ column. Afterwards the extract was washed with 20% MeOH and eluted with 50% and 100% methanol (2 CV). The latter fractions were

combined and concentrated under reduced pressure. This procedure was the same for agar and liquid extracts. The dried samples were dissolved in MeOH to a concentration of 50 μg/mL and analyzed by HR-UPLC-MS/MS.

**Reporting summary**. Further information on research design is available in the Nature Portfolio Reporting Summary linked to this article.

## Data availability
Supplementary Information contains details of experimental and analytical data. Supplementary Data 1, 2 contain details and summary of the literature data.

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

## Acknowledgements

This study was funded by the German Research Foundation (DFG, Deutsche For-schungsgemeinschaft) under Project-ID 239748522 – CRC 1127 (project A6). We are grateful for the financial support from the Agence Nationale de la Recherche (ANR, ANR-17-CE07-0051-01) and the German Research Foundation (DFG, BE 4799/3-1). This work was also supported by National Research Foundation of Korea (NRF) grants funded by the Korean government (MSIT; grant numbers 2019R1A5A2027340, 2021R1A2C2007937, and 2021R1I1A1A0104960613) and the European Research Council (ERC-CoG 771349). K.A.J. and K.H.B were supported by the Villum Foun-dation (Young Investigator Programme, project no. 15560) and the Carlsberg Foun-dation (Distinguished Associate Professor Fellowship, project no. CF17- 0248). We also thank staff and field assistants from the Binatang Research Center and local communities in Yawan, as well as the Conservation and Environment Protection Authority (CEPA) of Papua New Guinea for assistance with research permits and export permits.

## Author contributions

E.S., K.J., M.P., C.B. designed research. E.S., S.U., M.d.K., K.H.K.: performed the experiments. E.S., S.U., M.D., M.d.K., K.H.B., K.H.K., C.B.: analyzed data. E.S., S.U., M.D., K.H.B., K.H.K., C.B.: generated the figures. E.S., C.B., K.J., M.P.: wrote the manuscript with input from all authors.

## Funding

## Competing interests

The authors declare no competing interests.
