## [Peer Review File · Communications Chemistry]

Reviewers' comments:

Reviewer #1 (Remarks to the Author):

The authors discovered new macrotermycin derivatives via GNPS and verified the biosynthesis and regulatory mechanism of the compounds by a large number of experiments. They isolated and structurally characterized the compounds. The data was explained in a clear and logical manner and the scientific story is complete and interesting. I only have a few comments.

1. We suggest re-sequencing the genome of *Amycolatopsis* sp. M39 by joint use of second generation (Illumina) and third generation (e.g. Nanopore), to harvest the complete genome without the need to deduce the *mte* gene cluster
2. the resolution of Figure 2 is low.
3. Some NMR spectrum lack the associated ^{13}C spectrum, like Figure 35 and 36, lacking the ^{13}C spectrum in HMBC and HSQC. In addition, the attribution format of NMR spectrum should be unified. Please add the related spectrum and note other NMR spectrum existing related minor mistakes.
4. There are some typos and formatting errors in the manuscript (e.g. table in Figure 2B — others replaced other; the representation of optical rotation is wrong). Please proofread the manuscript and correct them.
5. The molecular weight of the MS2-based spectral network is retained 3 digits after the decimal point. The m/z is marked as 489.26 in Figure 6B for macrotermycin C but as 489.258 in article sentence.

Reviewer #2 (Remarks to the Author):

Seibel et al. reanalyzed the BGC of beta-amino acid-containing polyketides using existing genomic data. They obtained useful knowledge about them, and newly isolated and structurally determined an analog of ciromicin A. The paper is very interesting and I judge it worthy of publication in *Communication Chemistry*. Please consider revising the following points.

1. The current title does not fully reflect the first half of the content of this paper. Please revise it to something more suitable.
2. In Introduction. I find it very difficult to understand the contents of Figure 1 from the introduction alone, Figures S1 and S2 seem to contain useful information, but I don't think they are well explained in the text. If possible, please add more information to Figure 1 and add an explanation in the introduction that is appropriate for the introduction of this paper.
3. About Figure 2A. Reading the text and Figure Legend, it is not clear what information Figure 2A contains and how it helps to develop the logic. Please add more detailed explanation to the text.
4. In Figure 3. Please chemically correct the axial and the equatorial of UDP-N-

acetylglucosamine and its precursors. Correctly draw the absolute configuration of Ciromicin A at C-18.

5. In "Isolation of predicted macrolactam scaffolds". Please explain the results of the GNPS analysis in more detail. There is no data to support the configuration proposed in Figure S26. J-based configuration analysis method should be applied at a minimum.

6. In Figure 8B. I'm not sure how these expression vectors are configured. The figure is particularly small and unclear as to how the genes are introduced. Please add an explanation either in the text or in the supporting text.

7. In Figure 8EF. Both the text and the figure are very confusing as to which intermediates are being produced and to what extent. Please revise this section by adding the structures of the presumed intermediates so that the reader can clearly see what has been accomplished in this section. In light of this revision, I expect the DISCUSSION section to be written more clearly as well.

Reviewer #3 (Remarks to the Author):

The manuscript by Beemelmans and coworkers maps the distribution of putative macrolactam biosynthetic gene clusters across the actinobacterial phylum by using macrolactam-specific query sequences as bioinformatic bait (e.g., amino acid precursor biosynthesis; PKS machinery). This interesting analysis nicely shows the broad distribution of such pathways and gives statistical insights, e.g., concerning the frequency of the different amino acid building blocks. Building on these data, the authors isolate three new macrotermycin analogs from *Amycolatopsis* strain M39 (along with some corrections regarding macrotermycin biosynthetic assembly and stereostructure) and discover two new ciromicin A producers. Finally, they succeed in heterologous expression of the *asa* biosynthetic gene cluster in *S. albus*, thereby firmly connecting the pathway to macrolactam production and developing a tool for future macrolactam pathway engineering. Overall, this is a highly interesting paper that compiles a large number of new insights into macrolactam distribution and assembly. The manuscript is well written and nicely illustrated with graphics. Therefore, I suggest acceptance of this work, with only some minor issues to be addressed:

- Page 2, below Figure 1: ',... possess a or beta-alanine (beta-Ala) moiety.' There seems to be something missing in this sentence. Or should ',a' be ',alpha'?
- Page 5, new subchapter: maybe the authors should add 1-2 sentences as to why they chose to focus on *Amycolatopsis* sp.
- Page 6, ff: I suggest the authors add a graphical representation of the biosynthetic assembly to be expected in the *asa/pac* cluster. This would make this subsection much easier to read. In addition, intermediate III is later mentioned in the text (detected recombinant product), and it would thus be nice to have it depicted once.
- Figure 7:
 - the authors should not change the drawings of the macrotermycin structure when presenting

NMR interactions versus just the structure alone; maybe keep it as in the NMR data representations, as this would be consistent with the rest of the paper.

- There seems to be an error next to compounds E and F, where the substituent at C-18 should not be ,O' but rather ,CH3'.

- Can the authors propose the formation of compound G from the monocyclic macrolactam precursors?

• In the last sentence above the subchapter ,Heterologous expression of the asa BGC': should C-9 not rather be C-7?

Point-to-point response

	Reviewer 1 - Revisions	Possible response
2	resolution of Figure 2 is low	We included now a vector scalable figure to improve the resolution.
3	Some NMR spectrum lack the associated ¹³ C spectrum, like Figure S35 and S36, lacking the ¹³ C spectrum in HMBC and HSQC	We thank the reviewer for the critical reading. As noted in the NMR tables of macrotermycins, the assignments of the ¹³ C chemical shifts are based on ¹ H- ¹ H COSY, gHSQC, TOCSY, and gHMBC experiments. We acknowledge that analysis based on 2D NMR is accompanied by broader ¹³ C NMR signal compared to 1D experiments, but was necessary due to low amount of isolated material (<0.8 mg). We would like to point out that it was possible to unambiguously assign all chemicals shifts from the data set and that usage of 2D NMR is well accepted in the natural product chemistry fields for those compounds with low production titers.
4	attribution format of NMR spectrum should be unified	We have unified the legend and formatting related to all NMR spectra
5	There are some typos and formatting errors in the manuscript (e.g. lable in Figure 2B — others replaced other; the representation of optical rotation is wrong). Please proofread the manuscript and correct them.	We are thankful for the comments and apologize for the mistakes. We have corrected and proofread the manuscript more carefully.
6	The molecular weight of the MS2-based spectral network is retained 3 digits after the decimal point. The m/z is marked as 489.26 in Figure 6B for macrotermycin C but as 489.258 in article sentence.	We thank the reviewer for the detailed reading. We have unified the m/z values to 3 digits.
7	If possible, re-sequencing the genome of Amycolatopsis sp. M39 for a new hybrid genome assembly to harvest the complete genome without the need to deduce the mte gene cluster	We have indeed sequenced the genome of M39 overall three times using short read technologies and in addition now Nanopore sequencing. As anticipated the mte cluster of M39 was identified as deduced from short read sequencing and highly similar to the cluster of Amycolatopsis rubida DSM44637, which is likely the same species as M39. Due to the additional experiments (Nanopore sequencing, assembly, annotation and alignment, the contribution of Martinus de Kruijff has been acknowledged.

	Reviewer 2 - Revisions	Possible response
1	The current title does not fully reflect the first half of the content of this paper. Please revise it to something more suitable.	Indeed, we have revised the title to more accurately reflect the full content of the manuscript. "Genome mining for macrolactam-encoding gene clusters allowed for the network-guided isolation of β -amino acid-containing cyclic derivatives and heterologous production of ciromicin A " Thank you for pointing this out!
2	Introduction. I find it very difficult to understand the contents of Figure 1 from the introduction alone, Figures S1 and S2 seem to contain useful information, but I don't think they are well explained in the text. If possible, please add more information to Figure 1 and add an explanation in the introduction that is appropriate for the introduction of this paper.	We thank the reviewer for the constructive advice. To improve the text, we revised Figure 1 and the corresponding text.
3	Figure 2A. Reading the text and Figure Legend, it is not clear what information Figure 2A contains and how it helps to develop the logic. Please add more detailed explanation to the text.	We acknowledge the critical advice. We have added additional explanations in the text and a new Figure to give the reader a better story line.
	Figure 3. Please chemically correct the axial and the equatorial of UDP-N-acetylglucosamine and its precursors. Correctly draw the absolute configuration of Ciromicin A at C-18.	Thank you for critical viewing. All structures were corrected.
4	In "Isolation of predicted macrolactam scaffolds". Please explain the results of the GNPS analysis in more detail.	More information on the GNPS analysis were added.
5	There is no data to support the configuration proposed in Figure S26. J-based configuration analysis method should be applied at a minimum.	Coupling constant values have been added in the elucidation part discussion.
6	Figure 8B. I'm not sure how these expression vectors are configured. The figure is particularly small and unclear as to how the genes are introduced. Please add an explanation either in the text or in the supporting text.	A chapter for the description of vector construction was added in the method section of the manuscript and a more detailed vector map was added in the supporting items file (Figure S55).
7	Figure 8EF. Both the text and the figure are very confusing as to which intermediates are being produced and to what extent. Please revise this section by adding the structures of the presumed intermediates so that the reader can clearly see what has been accomplished in this section. In light of this revision, I expect the DISCUSSION section to be written more clearly as well.	We apologize that the graphical depiction was not clear. The corresponding figure was inserted for better understanding and the intermediates were clearly highlighted in all figures and referenced in the text.

	Reviewer 3 - Revisions	Possible response
8	Page 2, below Figure 1: ',... possess a or beta-alanine (beta-Ala) moiety.' There seems to be something missing in this sentence. Or should ',a' be ',alpha'?	The sentence has been corrected.
9	Page 5, new subchapter: maybe the authors should add 1-2 sentences as to why they chose to focus on Amycolatopsis sp.	We clarified in the text that the main purpose was the re-evaluation of the BGC proposed for macrotermycin biosynthesis in Amycolatopsis sp. M39 and the closer analysis of the organization and regulation of 4macrolactam BGCs. Also, Amycolatopsis showed the highest amount of detected BGCs (besides Streptomyces), giving us enough data for the cluster comparison.
	Page 6, ff: I suggest the authors add a graphical representation of the biosynthetic assembly to be expected in the asa/pac cluster. This would make this subsection much easier to read. In addition, intermediate III is later mentioned in the text (detected recombinant product), and it would thus be nice to have it depicted once.	We apologize that the graphical depiction was not clear. The corresponding figure was inserted for better understanding and the intermediates were clearly highlighted in all figures and referenced in the text.
10	Figure 7: - the authors should not change the drawings of the macrotermycin structure when presenting NMR interactions versus just the structure alone; maybe keep it as in the NMR data representations, as this would be consistent with the rest of the paper.	Structures with 2D NMR correlations have been corrected to keep the same representations in all the paper.
11	Figure 7: There seems to be an error next to compounds E and F, where the substituent at C-18 should not be ',O' but rather ',CH3'.	Structures have been corrected.
12	Figure 7: Can the authors propose the formation of compound G from the monocyclic macrolactam precursors?	A mechanism for the formation of macrotermycin G was added to Figure 8.
13	In the last sentence above the subchapter ',Heterologous expression of the asa BGC': should C-9 not rather be C-7?	We thank the reviewer for critical reading. In ciromicin, it is C-9 and in macrotermyin C-7. The sentence has been clarified.

REVIEWERS' COMMENTS:

Reviewer #1 (Remarks to the Author):

The authors have extensively revised the manuscript. I think it is now qualified to be published in communications chemistry.

Reviewer #2 (Remarks to the Author):

I have judged that the requested revisions have been appropriately addressed. I am confident that this revision will significantly improve the quality of the paper and better convey the impact of this paper to our readers.